# Soil organic carbon, physical fractions of the macro-organic matter, and soil stability relationship in lacustrine soils under banana crop

Tatiana Rondon[1,2]*, Rosa Mary Hernandez[3], Manuel Guzman[2]

**1** Facultad de Agronomía, Universidad Central de Venezuela (UCV), Maracay, Aragua, Venezuela, **2** Corporación Colombiana de Investigación Agropecuaria (AGROSAVIA), Centro de Investigación La Selva, Rionegro, Antioquia, Colombia, **3** Laboratorio de Biogeoquímica, Instituto de Estudios Científicos y Tecnológicos (IDECYT), Universidad Nacional Experimental Simón Rodríguez (UNESR), Altos de El Cují, Miranda, Venezuela

* trondon@agrosavia.co

**Data Availability Statement:** All relevant data are within the paper and on Figshare via a http://dx.doi.org/10.6084/m9.figshare.13507050.

## Abstract

Banana is a staple food and a major export commodity in the tropics. However, banana production systems are affected by the plant-soil relationships, where properties such as quality and quantity of soil organic matter play an important role in the dynamics of soil physical properties. In order to evaluate the effect of the soil organic carbon (SOC) content and its distribution in the water-stable of soil aggregates (WAS), and the physical fractions of the macro-organic matter, a study was conducted in lacustrine soils under Banana cv. 'Grand Nain' in Venezuela. Soil sampling was carried out in two batches differentiated by their textural class and crop production. A completely randomized design under a directed random sampling technique was carried out. In each condition, 12 composite samples were taken at depths 0–5 and 5–10 cm, respectively. WAS were separated into micro (< 250 μm) and macroaggregates (> 250 μm). Also, physical fractionation by size-density of the macro-organic matter into light (LF), intermediate (IF), and heavy (HF) fraction using a silica gel solution, and SOC, were determined and correlated with banana yield and other agronomic traits. A major proportion of aggregates > 250 μm were found in both conditions and depths. Organic Carbon within soil aggregates ranged between 29.7 and 35.3 g kg⁻¹. The HF was superior to IF and LF; however, its C content was higher in the LF. The results allow inferring that the stability conferred to these soils is primarily associated with the presence of the snail, which shares the same size as the aggregates studied. High yields are associated with high C content in stable aggregates, as well as in the most labile fractions of macro-organic matter. These results highlight the importance of the use of organic fertilizers less recalcitrant as a strategy for sustainable management of banana cultivation.

**Funding:** This work was funded by the Ministry of Science and Technology of Venezuela (www. mincyt.gob.ve), through the Organic Law of Science, Technology, and Innovation (LOCTI, Venezuela) grant 07-01-33-01, project: "Plan de fertilización integral de banano, Musa AAA para el aprovechamiento de las fuentes orgánicas en un suelo lacustrino de la depresión del Lago de Valencia". Finca Agropecuaria Punta Larga (www. puntalarga.net) paid for local field workers and reactive chemicals. The funders had no role in study design, data collection and analysis, decision to publish, or preparation of the manuscript.

**Competing interests:** The authors have declared that no competing interests exist.

## Introduction

Agricultural land use and cropping systems have promoted an accelerated degradation of soils with important changes in their chemical, physical, and biological dynamics. In this scenario, continuous soil uses without proper agricultural practices affect the sustainability of cropping systems and will be at risk in the mid-term [1–3]. Once the soil environment has interfered, the availability of nutrients such as nitrogen and phosphorus decreases, as well as the soil organic carbon (SOC) contents [4, 5], even when the soil management carried out does not involve changes in the vegetation cover [6].

A stable soil structure is the most desirable soil characteristic for sustaining agricultural productivity and preserving environmental quality [7]. Several soil properties are dependent on the dynamics of soil structure, which is usually defined as the aggregates' ability to resist the separation and dispersion under soil moisture, anthropic action, or other external factors. The aggregates are primarily formed by physical processes. Meanwhile, biological and chemical processes and their interaction are mainly responsible for their stabilization [8, 9]. Primary soil particles (sand, silt, and clay) are held together firmly, so they do not disperse in water by cementing agents such as soil organic matter (SOM) contents, quantity and types of clay particles, iron and aluminum hydroxides, and others [10, 11]. However, clay flocculation is a prerequisite for soil aggregation [12]. Soil aggregates are an important component of soil structure due to their importance in the protection of soil loss by erosion [13] and degradation [7]. In addition, aggregate formation in soils is an important process in SOC stabilization [14, 15], isolating it from decomposers [16]. Arshad and Coen [17] proposed aggregate stability as one of the soil physical properties that serve as an indicator of soil quality. According to Tisdall and Oades [18], aggregates are classified as microaggregates ($< 250$ μm) and macroaggregates ($> 250$ μm). The stability of soil macro and microaggregates are used to estimated soil erosion, erodibility, and surface runoff [19–21].

Ma *et al.* [22] indicated that it is necessary to understand the relationship between water-stable aggregates and soil fertility properties since macroaggregates are destroyed easily with cultivation but the organic matter within is decomposed partially. Biological soil properties are an important fraction of soil quality, and are maintained or improved through conservation agriculture farming practices causing positive changes in SOC accumulation and microbial activity [23]. The SOM has an important role as source and sink because of its interaction with several soil variables such as soil aeration, moisture retention, nitrogen fixation, and aggregate formation [24–26]. Chien *et al.* [27] indicated the influence of SOM over different soil processes and the importance to study the quantity and type of SOM.

In addition, the study of carbon pools in the soil aggregates allows knowing the dynamics of soil carbon sequestration and mineralization in the aggregates [28], as well as its influences on crop development and yields [29]. Several studies suggest a fractionation of SOM as an effective procedure to obtain more information on SOM types and their distribution in the soil matrix [30–32]. Techniques have been developed to measure the size and turnover of SOM pools and used to separate them into labile and recalcitrant pools. This classification is based on the relative susceptibility to biological decomposition [31, 33]. These methods rely on chemical, physical, or biological separation [34]. However, physical fractionation is the most used by the scientific community for being considered less destructive than chemical methods, and since it relates better to the structure and function of SOM *in situ* [35, 36]. Physical fractionation has been applied to determine the association of SOM with primary particles, and to quantify the amount of particulate organic matter between and within soil aggregates [37, 38]. Fractionation by density is based on the submersion of soil samples into inorganic salt solutions with a specific density between 1.6 to 2.2 g cm$^{-3}$

and limited sample dispersion [39]. This process separates the light and heavy fractions, where the light is considered more labile and with a density lower than the soil minerals. Meanwhile, the heavy fraction is assumed to be enriched in decomposition products stabilized onto the surface of clay or silt particles [31] or within soil microaggregates, making it more resistant to microbial degradation and with a higher specific density due to its intimate association with soil minerals.

In the present study, we investigated how the SOM, focused on the role played by their different physical fractions, affects the physical properties of lacustrine soils and banana productivity, which may have implications on the management and conservation of soils with this condition. To do this, we i) classified water-stables of soil aggregates, ii) fractioned SOM using a size-density method, iii) determined the C contents in each fraction, and iv) analyzed the relationship among SOM, stability, and banana yield. This study hypothesized that the relationships between the SOM management and C concentrations in the different SOM pools change soil physical properties dynamics, and affect the vigor and productivity of banana crops.

## Materials and methods

### Site description

Soil samples were taken in January 2009 in a commercial banana plantation with cv. 'Grand Nain' (*Musa* AAA, subgroup Cavendish) located at Lake Valencia basin, Venezuela (10° 8'33"N, 67°35'3"W), chosen for this study due to its high productivity standards focused on exportation markets. Lake Valencia basin accounts for more than 17,000 hectares in size under agriculture, mainly by field crops such as banana, sugarcane, and short-cycle vegetables. Despite the intensive use of agrochemicals and high-cost techniques in this area, a considerable reduction in productivity has been observed possibly due to the change and accelerated deterioration of the physical, chemical, and biological properties of the soil.

The sampling sites were in lacustrine soils characterized by a high-water table and the presence of a carbonitic soil layer with abundant snail shells and more than 40% of $CaCO_3$, placed in a Quaternary terrace. Altitude fluctuated between 410 and 440 m a.s.l. in the sampling area with a soil slope of $\leq$ 3%. This area is classified as Tropical Dry (Aw) climate according to the Köppen classification system. Long-term climate data indicated an annual precipitation average of 1028 mm, concentrated between mid-May and October with a unimodal pattern. The annual temperature average ranges from 17.5 and 28.0°C, evaporation of 1500 mm, and relative air humidity of 67%. The soils under study were classified as Entisol, with moderate well-drainage to well-drainage.

The banana plantation studied presents marked differences in fruit yield among its plots and were divided in two levels (Condition 1 and 2), according to an arbitrary, yet parsimonious, principle: textural class and yield. This contrasting principle combined the average banana fruit yield in Venezuela (~ 28 t ha$^{-1}$), information given by the farmers about plots behavior, and productivity parameters such as potential production and plant vigor.

An initial chemical and physical profile of soils studied were reported by Sapucky [40] and Cruz [41], obtained from soil pits (60 cm$^3$) (Table 1). Soils were classified as Mollic Ustifluvents, sandy loam, mixed, calcareous, isohyperthermic (Condition 1) and Typic Ustifluvents, fine-loamy, calcareous, isohyperthermic (Condition 2), according to USDA soil taxonomy system [42]. Sapucky [40] indicated that soil physical variables such as textural class, bulk density, porosity, moisture retention capacity were related to banana yield and productivity in these soils.

**Table 1. Soil properties of sampling sites studied.**

| | Condition 1 | | Condition 2 | |
|---|---|---|---|---|
| **Property** | **0–5 cm** | **5–10 cm** | **0–5 cm** | **5–10 cm** |
| **Textural class** | Silt loam | Silt loam | Loam | Loam |
| Sand (%) | 8.80 | 9.10 | 7.20 | 7.90 |
| Silt (%) | 73.3 | 75.2 | 69.0 | 68.3 |
| Clay (%) | 18.8 | 15.7 | 23.1 | 24.5 |
| Bulk density (g cm-3) | 1.15 | 1.10 | 1.00 | 1.00 |
| Soil pH (1:1) | 7.73 | 7.77 | 7.84 | 7.78 |
| Soil electrical conductivity (dS m-1) | 0.87 | 0.78 | 1.62 | 2.10 |
| Total soil porosity (%) | 57.7 | 58.0 | 60.1 | 59.0 |
| Soil Organic Matter (%) | 4.42 | 4.92 | 5.91 | 4.89 |
| Saturated Hydraulic Conductivity (Ks) (cm h-1) | 1.69 | 3.89 | 6.56 | 6.55 |
| Penetration resistance (kPa cm-1) | 0.41 | 0.35 | 0.38 | 0.32 |

Note: adapted from Sapucky [41] and Cruz [42]

## Soil sampling procedure and experimental design

Representative plots of both conditions were selected for this study, following homogeneous selection criteria such as cultivar, a second cycle of plantation, agricultural practices, and physical and chemical soil conditions. A completely randomized design was used in combination with a simple random sampling technique with 6 replications, targeted in plants at the flowering stage. Each experimental unit or plot size was of 400 m$^2$. At 'Condition 1' and 'Condition 2', any litter layer was removed before soil sampling, and sampling was done in the base of the parent plant and close to the following sucker. All plants selected for soil sampling were at the initial flowering stage.

Soils cores were taken with a shovel from 2 depth increments: 0–5 cm and 5–10 cm. In both samplings, 3 kg of composite soil samples were collected for each depth. Each composite sample derived from 12 sub-samples collected in different points on the plot; all were well mixed in a bucket. From the soil extracted with the shovel, only the central part was recovered to avid collecting aggregates disturbed by sampling. A hand-held GPS device with 0.5 m precision was used to georeferenced the location of sampling points.

## Soil and crop variables evaluated

Composite soil samples were air-dried, visible plant roots removed and grounded to pass through a 2.0-mm sieve. These grounded soil samples were stored in hermetic plastic bags until analyses. All laboratory analyses were replicated three times for each sample. The water-stables of soil aggregates (WSA) were determined using an Eijkelkamp wet sieving apparatus (Eijkelkamp Agrisearch Equipment, Glesbeek, The Netherlands) following the description developed by Florentino [43] with modifications. Sub-samples of 20 g of 1–2 mm air-dried aggregates were pre-wetted by capillary action using a paper napkin with distilled water, by 5 minutes. Later, pre-wetted sub-samples were put into the Eijkelkamp apparatus and wet sieving by 3 minutes at a constant speed. The amount of WSA was determined based on soil material kept on the 250 μm mesh, classified as soil macroaggregates (> 250 μm) and microaggregates (< 250 μm), respectively. The macro and microaggregates were dry in an oven at 105°C by 24 h and weighted.

Size-density fractionation of SOM was conducted using a silica suspension (LUDOX) adjusted to 1.37 g cm$^{-3}$ as described by Meijboom *et al.* [44] with modifications (Fig 1). In

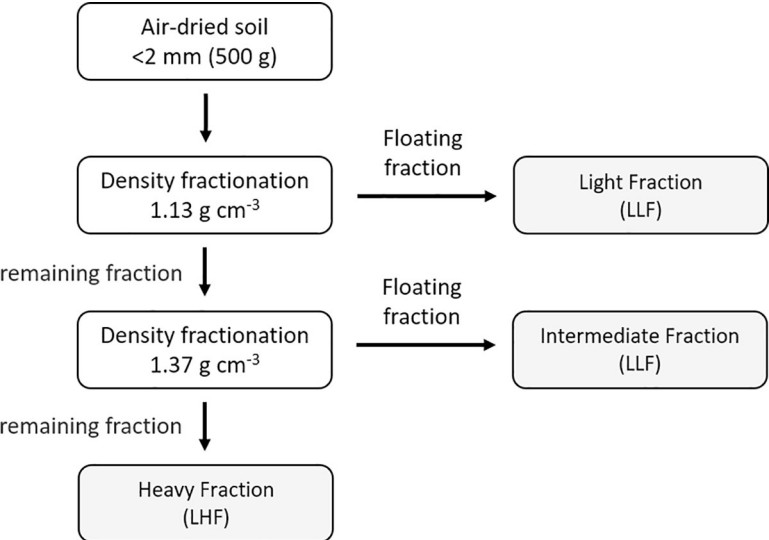

**Fig 1. Schematic representation of size-density fractionation of SOM.**

short, 500 g of air-dried 2 mm sieved soil was weighed into 3 L beaker, 2 L of deionized water was added and vigorously shaken, with the aim to separate particulate organic matter from soil mineral fraction. The suspension was filtered through superimposed sieves of 250 and 150 μm, respectively located at top and bottom, and washed carefully to separate by size. A jet of water was applied into soil material retained on the sieves, washed, and swirled to separate macro-organic matter from mineral fraction by decanting. Swirling and decanting were repeated several times until no floating materials remained using the 150 μm sieve. The organic material collected in the 150 μm sieve was submerged in the LUDOX suspension adjusted to 1.13 g cm$^{-3}$ by ten minutes. The floating fraction was collected and correspond to the light fraction (LF). The remaining fraction in the sieve was placed in LUDOX suspension adjusted to 1.37 g cm$^{-3}$ by ten minutes. The new floating fraction was an intermediate fraction (IF) and the non-floating fraction corresponded to the heavy fraction (HF). All three fractions collected were washed with deionized water and dried in an oven at 50°C until to constant weight.

Soil organic carbon (SOC) by each WSA (macro and microaggregate), and SOM fractions (LLF, LIF and LHF) were determined using wet digestion method according to Walkley-Black [45] method and modified by Heanes [46], using potassium dichromate ($K_2Cr_2O_7$) and concentrated sulfuric acid ($H_2SO_4$).

At harvesting, in the same plots where soil samples were collected, banana crop traits were recorded such as yield (Y, t ha$^{-1}$ year$^{-1}$), pseudostem diameter (PD), determined by the circumference at 100 cm from the ground; plant height (PH), from the base of pseudostem to the point of bunch emergence expressed in cm; number of hands per bunch (HB), and bunch weight (BW) expressed in kg.

## Data analyses

The normality of data was assessed using Shapiro-Wilk test ($p \leq 0.05$). A one-way analysis of variance (ANOVA) was used to analyze the data. Means were separated using Tukey test at $p \leq 0.05$ if there was a significant site effect. Correlation analysis by Pearson was used to determine the relationship between each pair of traits evaluated, using a statistical significance at the $\alpha < 0.05$ level.

## Results

### Water-stables of soil aggregates (WSA)

Significant differences in WSAs were observed between the conditions studied (Fig 2). In both examined soil depths, the proportion of > 250 μm aggregate fraction was higher in condition 2, indicating that these soils were composed mainly of macroaggregates. Water stable > 250 μm aggregates were 77.41% and 89.25% for condition 1 and 2 at 0–5 cm, respectively; meanwhile, 75.27 and 84.23% was found at 5–10 cm. In contrast, a higher proportion of < 250 μm aggregate was found in condition 1, with more than 20%. The < 250 μm aggregate size range was low in both depths, with values between 7.98 (condition 2, depth 0–5 cm) and 23.29% (condition 1, depth 5–10 cm). Both conditions were stable in structural terms; however, the condition 1 showed a higher percent of aggregates associated with the microaggregates (< 250 μm) than condition 2.

### Organic carbon contents within soil aggregates

OC contents within soil aggregates showed highly significant differences ($p \leq 0.01$) among conditions studied at depth and size-aggregates level (Fig 3). In soil aggregates superior to 250 μm, the OC was superior in the condition 1 to both soil depths, with values of 32.7 and 29.7 g kg$^{-1}$, to 0–5 and 5–10 cm, respectively. On the other hand, in aggregates < 250 μm the OC content was superior in the condition 2 with values of 35.3 and 32.3 g kg$^{-1}$, to 0–5 and 5–10 cm, respectively. In both conditions, a decrease was observed in the OC content at deeper soil layers.

### Size-density fractionation of macro-organic matter (MOM)

Physical fractionation of LF, IF, and HF is showed in Table 2. Highly significant differences ($p \leq 0.01$) for all fractions in both soil depths were observed. 'Condition 2' showed higher values than 'Condition 1' for all fractions at 0–5 and 5–10 cm. Across soil depths and conditions, values ranged between 0.06–0.49, 0.14–1.50, and 0.45–2.93 g kg$^{-1}$, to LF, IF and HF, respectively. In general, HF was superior to the other fractions, showing a fraction decreased in the order HF > IF > LF, for the soils studied.

### Soil organic carbon (SOC) contents and its distribution in SOM fractions

Significant differences in the C content in the different SOM fractions were found between conditions in both depths (Fig 4). The C content was generally lower at 5–10 depth, except in the light fraction at 0–5 cm. In both depths, the C content decreased across the fractions in the order LF > IM > HF. At 0–5 cm depth, mean values of 21.7, 18.2 and 10.9 g C kg$^{-1}$ soil were found to LF, IM, and HR, respectively, across conditions; meanwhile, at 5–10 cm depth values of Carbon (g kg$^{-1}$ soil) were 26.7 (LF), 14.1 (IF) and 8.1 (HF). Only in the LF, an increasing trend in C content was observed at deeper soil depths. IF and HF showed a decrease in C content for both conditions.

### Banana yield and agronomic variable evaluated

Yield and other agronomic variables showed a significant difference at 0.05 level among conditions studied (Table 3), where all evaluated variables were superior in condition 2. Yield average was 67.6 t ha$^{-1}$ year$^{-1}$ across the condition studied, where the 'Condition 2' had a higher yield than 'Condition 1', with 74.61 and 60.48 t ha$^{-1}$ year$^{-1}$, respectively, representing an increase in yield of 23.4%. Traits such as plant height, pseudostem diameter, hands per bunch and bunch weight are used as an indicator of vigor in a banana plantation. Plant height was 44

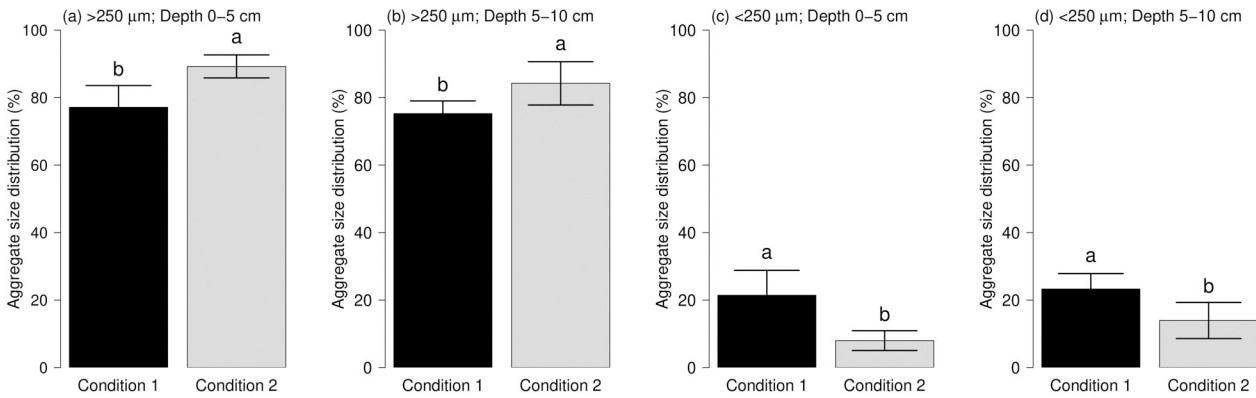

**Fig 2. Distribution by the size of water-stables aggregates at two soil depths in lacustrine soil studied.** Barplot (a) and (b) represent > 250 μm aggregates at 0–5 and 5–10 cm, respectively. Correspondingly, barplot (c) and (d) represent < 250 μm aggregates at 0–5 and 5–10 cm, respectively. Different letters indicate statistically significant differences at the 0.05 level by Tukey's post-hoc test.

cm superior in condition 2 than condition 1. Pseudostem diameter presented an average of 82.9 cm. The number of hands per bunch showed a difference of 1.85 between the conditions studied; meanwhile, the bunch weight ranged between 33.61 and 36.59 kg among conditions.

## Relationship among soil aggregates, size-fraction, and content of organic carbon, and agronomic traits

To identify the relationship between pairs of the variables studied, a Pearson correlation analysis was performed (Fig 5). The C content in the macroaggregates (CMA) was highly and negatively correlated with the CIF and CHF, with values of $r = -0.69$ and $r = -0.74$, respectively. However, to both variables, a positive correlation was found in the C content in the soil microaggregates (CMI). The C content in the different fractions of SOM: light (CLF), intermediate (CIF), and heavy (CHF) were evaluated. A high relationship between the different fractions of the SOM and the WSA was found. A high positive correlation ($p \leq 0.01$) was found among yield and PH, OC, HB, BW, PD, CLF, and CMA; meanwhile, a negative correlation was found among CIF, CHF, and CMI. PH also had a significant correlation with all variables studied, positive in all cases except for CIF, CHF, and CMI, indicating that these variables did not a

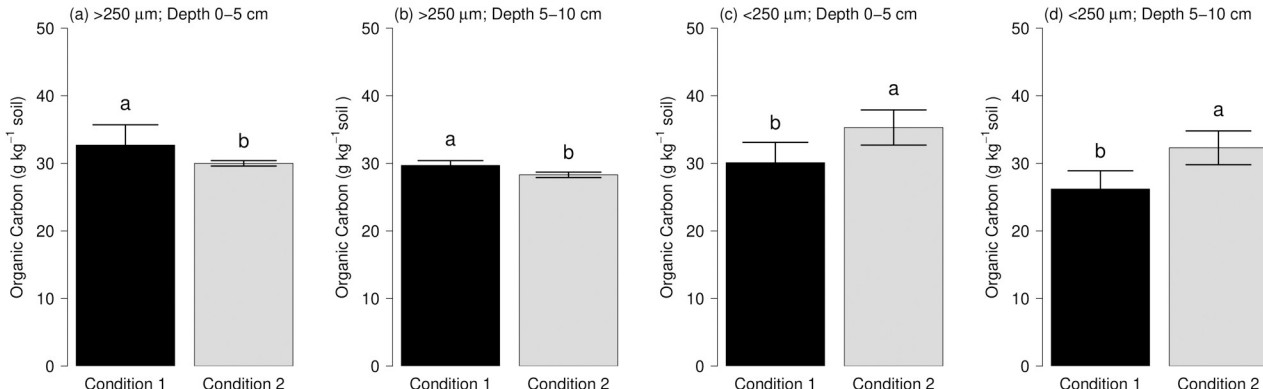

**Fig 3. Content of organic carbon associated with soil macro (> 250 μm) and microaggregated (< 250 μm) at two soil depths in lacustrine soil studied.** Barplot (a) and (b) represent > 250 μm aggregates at 0–5 and 5–10 cm, respectively. Correspondingly, barplot (c) and (d) represent < 250 μm aggregates at 0–5 and 5–10 cm, respectively. Different letters indicate statistically significant differences at the 0.05 level by Tukey's post-hoc test.

**Table 2. Distribution (g kg$^{-1}$) of macro-organic fractions studied.**

| Soil depth (cm) | | LF (g kg$^{-1}$) | IF (g kg$^{-1}$) | HF (g kg$^{-1}$) |
|---|---|---|---|---|
| 0–5 | Condition 1 | 0.15 ± 0.05 b | 0.52 ± 0.25 b | 1.27 ± 0.38 b |
| | Condition 2 | 0.44 ± 0.15 a | 1.50 ± 0.40 a | 2.93 ± 0.28 a |
| 5–10 | Condition 1 | 0.06 ± 0.03 b | 0.14 ± 0.05 b | 0.45 ± 0.10 b |
| | Condition 2 | 0.49 ± 0.17 a | 0.79 ± 0.22 a | 1.98 ± 0.25 a |

Different letters indicate statistically significant differences at the 0.01 level by Tukey's post-hoc test. Data expressed as mean ± SD.

positive contribution on yield. Significantly positive values ($p \leq 0.05$) were found between OC and HB, and among PD, BW, CLF and PH. Vigor traits such as PH, HB, PD, BW, and Y were correlated positively. In addition, a positive correlation was found between the CMA and all variables associated with the vigor of the plantation.

## Discussion

In the current study, the aggregate distribution shows that more than 70% of the soil was composed of macroaggregates (> 250 μm) in both conditions. Our results further reveal that condition 2 has a superior proportion of aggregates > 250 μm at depths of 0–5 and 5–10 cm. In the soil, the influence of the aggregate size over other factors such as water retention, and water and air movement are crucial. This allows establishing relations among pore distribution, erosion, water movement, gaseous exchange, and crop productivity [47, 48]. In the initial survey (Table 1), the total porosity ranged between 57.7 and 60.1%, being higher in the condition 2, as well as found in the soil macroaggregates distribution (Fig 2). This suggests a better aggregate stability of the soil. When the proportion of large to small aggregates increases, soil quality generally increases; however, this is highly dependent of SOM and biological activity in the soil.

The storage of SOC at soil aggregates differs from the behavior of WSA proportion. At condition 2, higher C contents were found in soil microaggregates (< 250 μm). In addition, a better structural stability and a SOC stable or physically protected to the action of external agents such as microorganisms, raindrop impact or anthropogenic action. The results suggest that the high content of SOM (5.91%) in this condition plays an important role as a cementing

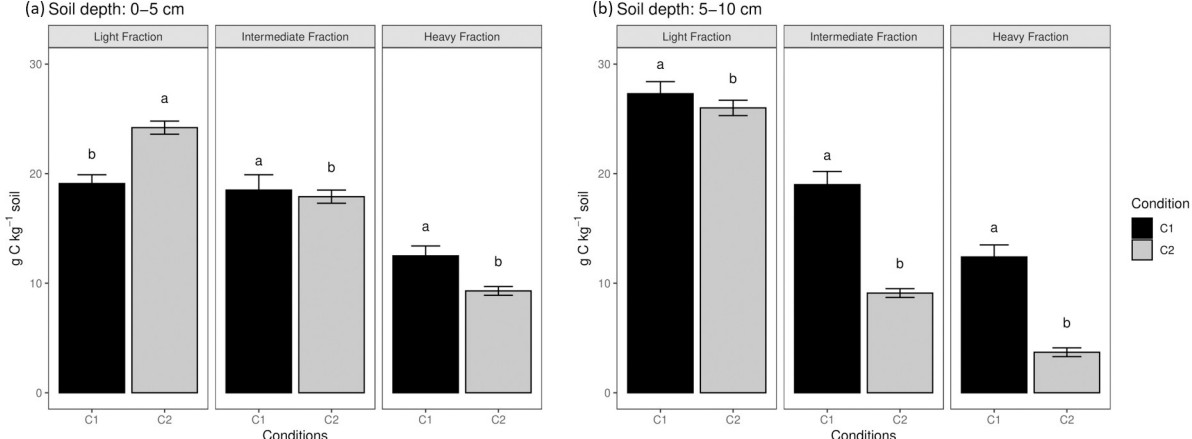

**Fig 4. Organic carbon (g kg$^{-1}$ soil) associated with the different macro-organic matter fractions.** Light, intermediate, and heavy fraction at (a) 0–5 and (b) 5–10 cm, respectively. Different letters indicate statistically significant differences at the 0.05 level by Tukey's post-hoc test.

**Table 3. Yield and agronomic variables evaluated in this study.**

|  | Y (t ha⁻¹ year⁻¹) | PH (cm) | PD (cm) | HB | BW (kg) |
|---|---|---|---|---|---|
| Condition 1 | 60.48 b | 282 b | 75.3 b | 9.5 b | 33.61 b |
| Condition 2 | 74.61 a | 326 a | 90.6 a | 11.4 a | 36.59 a |

Y: yield, PH: plant height, PD: pseudostem diameter, HB: hands per bunch, BW: bunch weight. Different letters indicate statistically significant differences at the 0.05 level by Tukey's post-hoc test

agent in the WSA and SOC. This matches the observations made by Tisdall and Oades [18] and Cabria *et al*. [49]. The soil aggregate stability may be under the influence of cementing or binding agents, which may be inorganic, organo-mineral association or organic, as is suggested by Tisdall and Oades [18]. High contents of SOC were found associated with recalcitrant or stable forms such as soil aggregates size lower than 250 μm, indicating a high activity of soil microorganisms, which are highly efficient in the decomposition of SOM. Our results further signify a high relation between SOC in aggregates with yield and vigor traits such as plant height and pseudostem diameter (Fig 5). In contrast, condition 1 shows lower SOC content, which may be a reason for lower structural stability. Also, this condition presents low values of OC associated with less recalcitrant form such as aggregate size greater than 250 μm, as a consequence of the deficient activity of soil microorganisms. Jastrow and Miller [50] suggest that the stability of microaggregates is influenced by the forces with which the particulate fraction of OM is able to suck clays, and inorganic and organic compounds, microbial residues, and organic colloids.

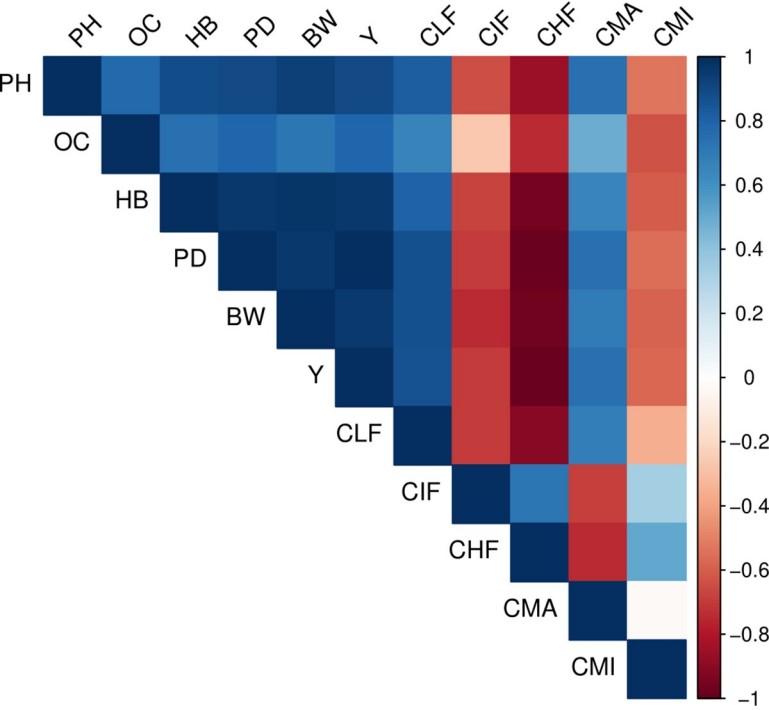

**Fig 5. Heatmap of correlation coefficient among soil aggregates, size-fraction and content of organic carbon, and agronomic traits.** Pearson correlation is showed between variables described in this study. Yield (Y), pseudostem diameter (PD), plant height (PH), number of hands per bunch (HB), bunch weight (BW), carbon content at light fraction (CLF), carbon content at intermediate fraction (CIF), carbon content at heavy fraction (CHF), carbon content at macroaggregates (CMA), carbon content at microaggregates (CMI).

The MOM fractionation by size and density suggest that MOM tends to be heavy due to a rapid transformation of organic residues (inputs) into humified forms. In both conditions, the distribution of macro-organic fraction follow the order HF > IF > LF, at the soil depths studied. Condition 2 shows a faster and more efficient decomposition of residues by the microorganisms. This result agrees with the report by Christensen [39], Zagal *et al.* [51] and Caballero [52], who indicate that the LF does not exceed 4 g kg$^{-1}$ in cultivated soils. Different studies also specify that the LF is sensitive to soil management practices [30, 53, 54]. This occurs because the LF is regarded as highly labile, composed mainly by organic matter that still does not form complexes with minerals of the soil [55]. In this sense, Caballero [52], using LUDOX as fraction separator, discover that the proportion of each fraction, is mainly due to the land-use and OC dynamics, in special at soils under crops by the microbial necromass contributions and the level of recalcitrant compounds.

The C content decreased across the MOM fractions in the order LF > IM > HF, but was associated mainly with the LF and IF, to both conditions and soil depths studied. These results suggest that the OC with more content in the MOM is unstable and easy to decompose. Six *et al.* [56] suggest a relation between the loss of organic C, N and structural stability with an increase of soil cultivation. A decreased of macroaggregates with high C content and an increase of microaggregates depleted in C is considered as one of the most important reasons for the loss of SOM [56–58]. The content of C in all fractions was correlated with the examined crop traits. However, high C contents in the recalcitrant fractions of SOM such as HF, showed a detrimental in the vigor of the studied banana plantation (Fig 5).

The complexity of MOM fractions varies according to its size; thus, smaller fractions are related to a higher rate of recalcitrance, complexity or resilience in the soil [59]. In general, banana plantations in Venezuela use as organic matter inputs from any tissues or leaf blades of the same crop, and organic compost as a complement. These inputs are labile and are quickly decomposed by the microorganisms, leading to more stable forms of MOM such as the IF and HF. Balesdent *et al.* [60] indicated the importance of the fractions of SOM, given its role on the C dynamics on cultivated soils, including fractions called 'inert' or humus, even organic residues at the same level of importance. With this, a comprehensive approach is created including SOM quality and the transformation of organic fractions that involves both the micro-environment and the substrate where the interactions occur. A balance of the fractions of SOM is important because it works as an indicator of 'organic composition', where physical, chemical and soil management cause modifications of SOM [61].

The soils studied are lacustrine, with a high content of snail shells in its structure. This characteristic was present in the micro and macroaggregates of the soil, as well in the physical fractions of SOM. Thus, the soil variables studied could be overestimated by the snail shell presence. This consideration is based on the presumption that the snail shell is part of the soil matrix. It is not considered as a soil aggregate even if it behaves similarly, regarding its conceding structure and porous space; but is neither considered as MOM. However, the analyses of the variables studied are based on weight and density, so its presence is necessarily considered.

## Conclusions

The distribution of aggregates and its stability were different between the soil conditions studied. The stability of the soil aggregates was higher under the 'Condition 2', due to a major proportion of soil macroaggregates (> 250 μm) at both studied depths. Also, this condition showed a high C content associated in aggregates < 250 μm, contributing to an increase of the OC stability, and improving the soil-plant system. The high C content at macroaggregates showed a positive effect on banana yield and other agronomic variables evaluated. The physical

fractions of SOM revealed a decrease in its proportions in the order HF > IF > LF. However, the dynamic of Carbon was inverted in the order CLF > CIF > CHF, facilitating the presence of Carbon into the labile fraction of MOM. The results indicate that the management of organic fertilization in these soils must be carried out with organic sources that promote high Carbon available levels in the labile soil fractions. Future research should focus in integrating approaches that combine chemical parameters in the soil and banana plants to evaluate this combination in the light of the soil organic matter.

## Acknowledgments

The authors thanks to the technical personnel of Laboratory of Soils Science of Universidad Central de Venezuela and Universidad Nacional Experimental Simón Rodríguez.

## Author Contributions

**Conceptualization:** Tatiana Rondon, Rosa Mary Hernandez.

**Data curation:** Tatiana Rondon.

**Formal analysis:** Tatiana Rondon, Manuel Guzman.

**Investigation:** Tatiana Rondon, Rosa Mary Hernandez, Manuel Guzman.

**Methodology:** Tatiana Rondon, Rosa Mary Hernandez.

**Supervision:** Rosa Mary Hernandez.

**Writing – original draft:** Tatiana Rondon, Manuel Guzman.

**Writing – review & editing:** Tatiana Rondon, Rosa Mary Hernandez, Manuel Guzman.

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
