## [Decision Letter · Decision Letter 0]

27 Apr 2021

PONE-D-20-41023

Soil organic carbon, physical fractions of the macro-organic matter, and soil stability relationship in lacustrine soils under banana crop

PLOS ONE

Dear Dr. Rondon 

Thank you for submitting your manuscript to PLOS ONE. After careful consideration, we feel that it has merit but does not fully meet PLOS ONE’s publication criteria as it currently stands. Therefore, we invite you to submit a revised version of the manuscript that addresses the points raised during the review process.

We look forward to receiving your revised manuscript.

Kind regards,

Tunira Bhadauria, Ph.D.

Academic Editor

PLOS ONE

Journal Requirements:

2)  We suggest you thoroughly copyedit your manuscript for language usage, spelling, and grammar. If you do not know anyone who can help you do this, you may wish to consider employing a professional scientific editing service.  

Reviewers' comments:

Reviewer's Responses to Questions

**Comments to the Author**

1. Is the manuscript technically sound, and do the data support the conclusions?

Reviewer #1: Partly

Reviewer #2: Yes

2. Has the statistical analysis been performed appropriately and rigorously? 

Reviewer #1: Yes

Reviewer #2: Yes

3. Have the authors made all data underlying the findings in their manuscript fully available?

Reviewer #1: No

Reviewer #2: Yes

4. Is the manuscript presented in an intelligible fashion and written in standard English?

Reviewer #1: Yes

Reviewer #2: Yes

5. Review Comments to the Author

Reviewer #1: 1. The author conducted a thorough and comprehensive laboratory and field study on Soil organic carbon, physical fractions of the macro-organic matter, and soil stability relationship in lacustrine soils under banana crop. The comprehensive data generated from this experiment will help other scientists working in the similar research area to address the same issue. However, I have identified some moderate level of issues in this manuscript (some editorial and some scientific) which will certainly help to improve the quality of the manuscript. My comments are intended to further improve the manuscript such that it addresses the questions asked properly. I would consider this manuscript for publication if the moderate changes are made and resubmitted. I strongly recommend that author should read the entire manuscript and do not rush on anything before final submission

I could not find any figures and strongly recommended for the inclusion of figures.

Reviewer #2: The present paper "Soil organic carbon............under Banana plantation " has got merit for publication. Very good study have been done with through knowledge of concept, methodology adopted and discussed in the paper. However at certain places the introduction part is lengthy and should be shortened, may be added in discussion. Besides following points need to be clearified before final decision:

1. Why condition 1 and condition 2 in the table is categorised as based on banana yield despite having two different soil type: sandy loam and Loam. It is the soil quality that influence yield not the yield influence the soil quality. Therefore, classification of condition 1 and 2 should be soil quality not yield.

2. Table 2 should be redrawn where coloumn 1 should have only two horizontal division :one for 0-5 cm soil and second for 5-10cm soil.

6. PLOS authors have the option to publish the peer review history of their article (what does this mean?). If published, this will include your full peer review and any attached files.

Reviewer #1: No

Reviewer #2: No

---

## [Author Response · Author response to Decision Letter 0]

3 Jun 2021

Reviewer #1: 

Thank you for comments. We considered all the specific comments to improve the grammar in the manuscript. Several sentences and words were re-word and highlighted using Track Charges. Figures were included in the recent version of the manuscript.

Reviewer #2:

Thank you for comments and suggestions. Introduction was revised and adjusted. In M&M, site description and soil sampling sections were improved to explain the criteria used to separate the banana plots studied. Table 2 was revised and adjusted according to the suggestion. 

General Comments:

*Abstract: revised and adjusted according to the suggestions.

*Introduction: Introduction was revised and adjusted, improving the grammar.

*Materials and Methods: sentences were revised and modified. Spacing between words were revised and adjusted. 

*Figures: figures were included in the main text, only for submission. Later, will be delated by the editor or authors.

Comments in the manuscript and responses. 

N° Comment Response

1- Comment: Change “biological and chemical” by “biochemical”; Response: Partially accepted. Was included “and their interaction”. We think that both processes independently contribute to the stabilization of the soil aggregate. 

2 Comment: Insert “it”; Response: Done. It was insert in the text.

3 Comment: SOM ??; Response: Corrected. The word was changed by “C”. the authors refer to the C content.

4 Comment: Very old study; Response: -

5 Comment: why can't say soil type division as sandy loam and loam,respectively; Response: Accepted. Redaction was improved. This study generated its hypothesis from the results obtained by Sapucky (2007, B.S. Thesis). This author evaluated basic chemical and physical properties, in the same study area, and indicated the relative importance of physical over chemical variables to explain banana productivity; and recommended a detailed SOM and its implication on soil physical properties. At ACORBAT meetings, Rey et al. (2006)* and Delgado et al. (2010)** indicated similar results on these soils.

6 Comment: Change “starting” by “initial”; Response: Correct. The word was changed.

7 Comment: Delete sentence; Response: Accepted. Sentence deleted.

8 Comment: Delete sentence; Response: Accepted. Sentence deleted.

9 Comment: Change “condition” by “sampling”; Response: Accepted and changed

10 Comment: Change by “for each”; Response: Accepted and changed

11 Comment: Italic; Response: Done. Corrected in all the text to consistency

12 Comment: Insert “method”; Response: Correct. The word was added. 

13 Comment: should have only two horizontal division .one for 0-5cm and second for 5-10 cm soil. Condition 2 should be written in parentheses.; Response: Accepted. Table was adjusted in the new version of the manuscript.

14 Comment: Change by “C content”; Response: Correct. The word was added.

15 Comment: Change by “C content”; Response: Correct. The word was added.

*Rey JC, Chacín M, Sapuky M, Núñez M, Martínez G, Rodríguez G, Espinoza J, Arturo M, Pocasangre L, Delgado E, Rosales F (2006) Land suitability for banana in Venezuelan soils and its relation to productivity. XVII Internacional Meeting ACORBAT: Banano un negocio sustentable. Joinville. Santa Catarina, Brasil. Nov 15–20. 

**Delgado EJ, Rey JC, Martínez G, Lobo-Luján D, Sapuky M, Chacin M, Nuñez E, Pocasangre L, Rosales F (2010). Soil properties that determine the vigor of locally produced banana (Musa AAA) in Venezuela. XIX Internacional Meeting ACORBAT Medellín, Colombia, nov 8–12

---

## [Editor Report · Decision Letter 1]

21 Jun 2021

Soil organic carbon, physical fractions of the macro-organic matter, and soil stability relationship in lacustrine soils under banana crop

PONE-D-20-41023R1

Dear Dr. RONDON

We’re pleased to inform you that your manuscript has been judged scientifically suitable for publication and will be formally accepted for publication once it meets all outstanding technical requirements.

Kind regards,

Tunira Bhadauria, Ph.D.

Academic Editor

PLOS ONE

Additional Editor Comments (optional):

The authors have revised the manuscript responding to each suggestion and incorporating these suggestions in the manuscript. Thus the manuscript has sufficient merit to be accepted for publication in the journal .
---

## [Editor Report · Acceptance letter]

1 Jul 2021

PONE-D-20-41023R1 

Soil organic carbon, physical fractions of the macro-organic matter, and soil stability relationship in lacustrine soils under banana crop 

Dear Dr. Rondon:

I'm pleased to inform you that your manuscript has been deemed suitable for publication in PLOS ONE. Congratulations! Your manuscript is now with our production department. 

Kind regards, 

on behalf of

Dr. Tunira Bhadauria 

Academic Editor

PLOS ONE